# Dynamic Models Supporting Personalised Chronic Disease Management through Healthcare Sensors with Interactive Process Mining

**DOI:** 10.3390/s20185330

**Published:** 2020-09-17

**Authors:** Zoe Valero-Ramon, Carlos Fernandez-Llatas, Bernardo Valdivieso, Vicente Traver

**Affiliations:** 1SABIEN-ITACA Universitat Politècnica de València, Camino de Vera S/N, 46022 Valencia, Spain; Carlos.FernandezLlatas@ki.se (C.F.-L.); vtraver@itaca.upv.es (V.T.); 2CLINTEC-Karolinska Institutet, 171 77 Solna, Sweden; 3Hospital Universitario y Politecnico La Fe, 46026 Valencia, Spain; valdivieso_ber@gva.es

**Keywords:** process mining, interactive, dynamic models, chronic diseases, obesity, hypertension, hyperglycemia, smart sensors

## Abstract

Rich streams of continuous data are available through Smart Sensors representing a unique opportunity to develop and analyse risk models in healthcare and extract knowledge from data. There is a niche for developing new algorithms, and visualisation and decision support tools to assist health professionals in chronic disease management incorporating data generated through smart sensors in a more precise and personalised manner. However, current understanding of risk models relies on static snapshots of health variables or measures, rather than ongoing and dynamic feedback loops of behaviour, considering changes and different states of patients and diseases. The rationale of this work is to introduce a new method for discovering dynamic risk models for chronic diseases, based on patients’ dynamic behaviour provided by health sensors, using Process Mining techniques. Results show the viability of this method, three dynamic models have been discovered for the chronic diseases hypertension, obesity, and diabetes, based on the dynamic behaviour of metabolic risk factors associated. This information would support health professionals to translate a one-fits-all current approach to treatments and care, to a personalised medicine strategy, that fits treatments built on patients’ unique behaviour thanks to dynamic risk modelling taking advantage of the amount data generated by smart sensors.

## 1. Introduction

The arrival of a new generation of mobile personal technologies, medicine sensors, and wearable sensors, has skyrocketed the quantity of data available nowadays [1]. This fact in combination with the massive introduction of Electronic Health Records (EHR) in medical systems has generated an enormous amount of information, the testimony of the patient’s passage along with the healthcare she/he receives. In this scenario, clinicians have not only the information collected within healthcare settings but also data coming from multiple sources, such as personal and environmental data, thanks to wearable, sensors, Internet of Things (IoT), mobile applications, or even social media.

These data could play an important role in the better management of chronic diseases also known as Non-Communicable Diseases (NCDs), as the concept of management includes detecting, screening, and treating. There are several definitions for NCDs, for example, the World Health Organization (WHO) defines them as diseases of long duration and generally slow progression [2], other authors [3], as having one or more of the following characteristics—they are permanent, leave residual disability, are caused by non-reversible pathological alteration, require special training of the patient for rehabilitation, or may be expected to require a long period of supervision, observation or care. Both definitions agreed in the fact of being of a long duration.

Chronic diseases, such as heart disease, stroke, cancer, chronic respiratory diseases and diabetes, are by far the leading cause of mortality in Europe, representing 77% of the total disease burden and 86% of all deaths [2]. The financial costs associated with treating chronic diseases are extremely high, and given that the average age of European populations is increasing, chronic diseases will continue to place an important pressure on national budgets [4]. Similarly, chronic diseases are among the most prevalent and costly health conditions in the United States. Nearly half (approximately 45%, or 133 million) of all Americans suffer from at least one chronic disease, having a great impact on health care costs [5]. Some of the chronic diseases with greater impact are coronary heart disease, stroke, many varieties of cancer, depression, diabetes, asthma, chronic obstructive pulmonary disease, or hypertension among others. Over 50 million people in Europe have more than one chronic disease, due to either random co-occurrence, possible shared underlying risk profile, or synergies in disease development [6]. Chronic conditions require ongoing management over a period of years or decades, so individuals’ behaviour should be taken into consideration. They are the result of a combination of genetic, physiological, environmental, and behavioural factors, that should be taken into consideration during their management, analysis, and treatment. However, data collected in EHR do not usually include such data, and consequently they could not be taken into account in order to manage chronic conditions.

Metabolic risk factors contribute to four key metabolic changes that increase the risk of suffering chronic diseases, these are raised blood pressure, overweight and obesity, hyperglycemia (high blood sugar levels), and hyperlipidemia (high levels of fat in the blood). In terms of attributable deaths, the leading metabolic risk factor globally is elevated blood pressure, to which 19% of global deaths are attributes [7], followed by overweight and obesity, and raised blood glucose [2]. Under these data, this study is focused on these three leading metabolic risk factors.

Hypertension, also known as high or raised blood pressure (BP), is a condition in which the blood vessels have persistently raised pressure. Based on WHO information, hypertension is a serious medical condition and can increase the risk of heart, brain, kidney, and other diseases. It is a major cause of premature death worldwide, and an estimated 1.13 billion people worldwide have hypertension [8]. Blood pressure is based in two numbers, systolic blood pressure (SBP) representing the pressure in blood vessels when the heart contracts or beats. And the diastolic blood pressure (DBP) representing the pressure in the vessels when the heart rests between beats. Hypertension is diagnosed if, when it is measured on two different days, the SBP readings on both days is 140 mmHg or more, and/or the DBP readings on both days is 90 mmHg or more or taking antihypertensive medication [9].

Obesity is another of the well-known chronic conditions; the WHO defines overweight and obesity as abnormal or excessive fat accumulation that may impair health. Body mass index (BMI) is a simple index of weight-for-height that is commonly used to classify overweight and obesity in adults. It is calculated by dividing a person’s weight in kilograms by the square of his/her height in meters (kg/m${}^{2}$) [10]. WHO also establishes a normal BMI range as 18.5 to 24.9, while a BMI greater than or equal to 25 kg/m${}^{2}$ and below 30 kg/m${}^{2}$ is considered to be overweight, and similarly, a BMI greater than or equal to 30 kg/m${}^{2}$ is classified as obese.

Finally, raised blood glucose or hyperglycemia is directly related to diabetes. Diabetes is a chronic, metabolic disease characterised by elevated levels of blood glucose, which leads over time to serious damage to the heart, blood vessels, eyes, kidneys, and nerves. Hyperglycemia is a common effect of uncontrolled diabetes and over time leads to serious damage to many of the body’s systems. The expected values for normal fasting blood glucose or fasting plasma glucose (FPG) concentration are between 70 mg/dL (3.9 mmol/L) and 100 mg/dL (5.6 mmol/L). When FPG is between 100 to 125 mg/dL (5.6 to 6.9 mmol/L) changes in lifestyle and monitoring glycemia are recommended. If FPG is 126 mg/dL (7 mmol/L) or higher on two separate tests, diabetes is diagnosed [11]. The difficulty with defining normality mirrors that of defining diagnostic cut-points for intermediate hyperglycemia that is, placing a specific cut-point on a continuous variable. Furthermore, other factors such as age, gender, and ethnicity are relevant to defining normality. Since there are insufficient data to accurately define normal glucose levels, the term ‘normoglycemia’ should be used for glucose levels associated with low risk of developing diabetes or cardiovascular disease, which is levels below those used to define intermediate hyperglycemia [12].

Measures for these three risk factors are capable of being collected with health sensors with a specific purpose, blood pressure, blood sugar and weight, joint with information collected from general purpose sensors, such as heart rate, sleep, activity, environmental data, coming from wearable, or personal mobile devices. The analysis of this amount of data could be used for creating new models for these three chronic conditions and deriving knowledge from data.

Since recently, the common fact of measuring physiological variables such as blood pressure or glucose levels was traditionally done by exams in a specialised health centre. Thanks to the development and introduction of a considerable set of sensors reading vital signs, such as blood pressure cuff, glucometer, heart rate monitor, including electrocardiograms, this situation has radically change, allowing patients to take their vital signs daily at home [13]. This has a double objective, on the one hand, patients are aware of their vital signs and can better manage their conditions. On the other hand, these data will significantly complement standard tests included in EHR.

For that, in recent years, much effort has been put in the design and development of smart sensors and mobile personal devices, that among others aim to improve people’s quality of life providing them with services and information about their health status and lifestyle. Also, with the explosion of the IoT, many applications managing data coming from sensors have become a reality in users’ daily life, allowing intelligent healthcare management, smart homes or intelligent environments. In the field of IoT in healthcare, devices could be used for remote monitoring, or emergency systems [14]. These medical devices for health monitoring may range from specific functions such as blood glucose, blood pressure, or heart rate, to more general ones for sleeping or activity monitoring. These specialised sensors could be used for collecting health status and used by physicians for monitoring patient’s health. This impact could be even great in the case of the management of chronic diseases. Thanks to sensors, patients’ healthcare can become more accessible, not only through the collection of physiological variables but also monitoring patient’s environment and lifestyle. At the end, all this information will support physicians in the development and delivery of more personalised treatment. It is clear that chronic diseases such as high blood pressure, diabetes, and obesity, which have a remarkable impact on socioeconomic aspects, could take advantage of the use of use mobile technologies and smart devices in the area of health.

Persons with chronic conditions are a large and growing segment of the population. Although chronic conditions are often associated with the older age population, evidence shows that 15 million of all deaths attributed to chronic diseases occur between the ages of 30 and 69 years [2]. This segment of the population has a long period for dealing with these diseases but they can also suppose an allied in the disease management, adopting and using a range of sensors or mobile personal devices. Despite sensors and related technologies already have some challenges, as precision, size, power consumption, communication and privacy, they provide valuable information for disease management, and ultimately for improving patients’ quality of life [13].

According to the above, this opens a new scene, where the challenge is not the lack of data, but how to exploit this huge amount of data. Moreover, this supposes an exceptional opportunity for creating new models and extracting knowledge from data. Consequently, an in-depth analysis of these data is paramount to obtain the necessary knowledge that allows, not only to improve the quality of the provided care but also better management of diseases and to move towards a patient-centred and value-based healthcare model, within the personalised medicine paradigm. Personalised medicine promises prediction, prevention, and treatment of illness that is targeted to individuals’ needs [15]. Furthermore, it is a demand within these new paradigms to analyse data in a dynamic and integrated way, instead of linear [15]. Another opportunity in the area is the development of new algorithms to support clinicians, new visualisation tools that show processes and models in an understandable way for health professionals and decision support tools, allowing more effective and precise management of diseases and treatments [16].

The main aim of this paper is to declare a novel method for representing dynamic risk models that characterise chronic diseases based on the evolution of the considered condition using Process Mining (PM) techniques. This model is a human-understandable graphical representation that could support healthcare stakeholders in comprehending their current awareness of the chronic disease processes, as it takes into consideration the disease’s variability over time and patient nature. Formerly, this work discusses the possibilities of the Interactive Process Mining paradigm for analysing data provided by health sensors to obtain new dynamic models for chronic conditions that consider patient’s behaviour over time, instead of static values, so we can infer real processes behind data for better management of chronic conditions. These models are called Interactive Process Indicators within the Interactive Process Mining paradigm, and their ultimate goal is the understanding, measurement, and optimisation of the processes associated with chronic diseases. This will allow health professionals to navigate behind the models and to discover the specificity of the processes associated with individuals.

This paper is not designed to be a comprehensive work of measures acquired through sensors but describes an example of chronic diseases modelling for three concrete metabolic risk factors, which variables could be easily obtained by sensors, but also included in a real EHR. The paper is structured as follows—Section 2 describes related work and the background, Section 3 materials and methods, Section 4 results and the last Section 5 describes conclusions and discussion, as well as future work.

## 2. Related Work

A treatment procedure that takes into account the patient’s unique behaviour, far from the *one size fits all* strategy, suits the personalised medicine paradigm. In this line, there is a necessity to detect what attitudes are followed by subjects after the idea of precision and personalised medicine. Treatments adequate to patients’ characteristics have a double impact, on one hand, they increase the effectiveness of the care pathways. And on the other hand, these treatments enhance the patient’s experience of care. In this scenario is important to find groups with similar characteristics and behaviours regarding the same condition, this will support better care delivery and maximise the process value [17]. Genetic sequencing or risk models could not be the unique variables to be considered in a precision medicine scenario [18], instead of this, health behaviour, mental health, social determinants, and patient preferences, coming from a variety of sensors and applications, should be also considered to achieve a full precision medicine-based care [19].

Notwithstanding, the way to approach knowledge extraction has been traditionally done by gathering data from clinicians and literature, and then these data have been used to develop health risk models in the preventive medicine concept. In this schema, risk models are statistical tools intended to offer *an individual probability for developing a future adverse outcome in a given period* [20]. Risk models are computed in a moment and have validity over time. Risk values, of an individual patient, play an important role in the decision taken by health professionals, who decide treatments delivered to patient depending on them.

The use of risk models introduces many benefits as they support and complement clinical reasoning and decision-making in medicine. However, within this approach risk models commit on static *snapshots* of variables, without considering any dynamic perspective. Traditionally, modelling, assessment, and management of chronic diseases have been done from a static and time-invariant set of concepts, definitions, and propositions, assuming a linear relationship between variables. But chronic diseases tend to be of long duration and are the result of a combination of genetic, physiological, environmental, and behavioural factors, that should be taken into consideration during management, analysis, and treatment. Moreover, these models do not take into account multiple data sources, as smart sensors or wearable, that could help in the better understanding of the diseases.

Furthermore, the temporal perspective of the clinical information is crucial for a complete awareness of a health process. Diseases are not static; they evolve towards different destinations, especially when talking about chronic health problems. For example, in Reference [21], the results suggested that optimal blood pressure management in children with chronic kidney disease (CKD) slows progression to end-stage renal disease and that works focused only on baseline blood pressure measurement may underestimate risk than using time-fixed blood pressure. In the same way, the human being is not static, a person is changing throughout her/his biography in age, lifestyle, socioeconomic status, or interconcurrent diseases.

The main benefits of using risk and prediction models in the healthcare domain are clear, however, since they are currently implemented, do not respond well to unexpected changes in patient’s conditions, as they suit standard conditions rather than unusual or unpredictable ones [22]. Individual differences cause great variances in the execution of models. In consequence, models of diseases should be dynamic, including disease variability and dependencies with other conditions, such as comorbidities, social conditions, or age.

To illustrate this problem, we can use one of the three conditions considered in this work, obesity. The excess weight derived from an obesity situation is a major risk factor to suffer other NCDs. Based on literature research, some of the comorbidities associated with overweight and obesity are cardiometabolic factors, including risk factors (hypertension, hyperlipidemia, and Type II Diabetes Mellitus), cardiovascular diseases, asthma, and musculoskeletal disorders [23,24,25]. When a patient is classified as *Obese* with a BMI greater than or equal to 30 kg/m${}^{2}$, the risk of comorbidities is considered as severe [24]. However, this is not only a question of patient’s current state, it is indeed more important to consider obesity onset, obesity evolution, weight fluctuations, duration of obesity (known as the time since BMI was first known to be at least 30 kg/m${}^{2}$), or even parental BMI to see comorbidities association and treatment [26,27]. Nevertheless, in real practise, if a patient decreases his/her weight, and, after a re-computation, achieves a *Normal BMI*, automatically all these risks disappear from the actual static care approach. In summary, the evolution of the risk model is not taken into account. Changes in the individual risk values are usually connected to behaviours, attitudes, and beliefs of patients. That means, people with the same disease and treated with the same treatment respond in different ways. Knowing the patient as an individual is key to select the best treatment for him or her [28].

Along this line, medicine sensors play a key role, as they provide measures of healthcare variables and lifestyle monitoring for personalised medicine [29]. There are some works in the literature about how sensors could help in the analysis and monitoring of chronic diseases, comprising the continuous collection of one or more vital signs, the processing, and the analysis to obtain medical parameters associated with the chronic disease under study. Studies go from state-of-the-art wearable sensors [29] or telemedicine platforms [14,30], to specific sub-parts of the system. Other work [31] proposes a new approach for translating IoT-based data into real-time clinical feedback. Most of the above works propose great examples of the application of medicine sensors to concrete situation, however they do not usually explore methods for analysing chronic diseases from a temporal and dynamic perspective that enable health stakeholders to obtain and understand individual healthcare processes associated to diseases, this is, an approach in which sensor data are transformed into new clinical evidence.

Moreover, there is an increased concern in discovering more precise stratification groups, that may allow to enhance care delivery and to augment the process value based on each group conditions [17]. There is a requirement not only to include health behaviour but also mental health, social determinants, and individual preferences to achieve a full precision medicine-based care [19]. Accordingly, there is a burden to stratify individuals built on their behaviour rather than in their disease [32]. In this regard, Process Mining technologies has been probed useful for creating individualised behaviour models [33].

The standardisation of the care process in medicine has been approached through Knowledge-Based Temporal Abstraction (KBTA or TA) [34]. Temporal Abstraction methods are thought to manage a switch from a qualitative time-stamped description of raw data to a qualitative interval-based representation of time series, with the main goal of abstracting high-level concepts from time-stamped data. In the literature, there are works approaching health processes with TA in some areas, such as the costs’ evaluation associated with Diabetes Mellitus [35], the prognosis of the risk for coronary heart disease [36], or for defining typical medial abstraction patterns [37]. These works tried to create an automatic summarising of patient’s current state based on patient’s data through temporal abstraction, nevertheless, most of the clinical variables (such as weight, blood pressure or blood glucose) have numerical values, and TA techniques are based on discrete labels, excluding important information from the analysis. In Reference [38], the author implemented a dual approach, Temporal Abstractions in combinations with Process Mining for blood pressure and temperature. Whereas Reference [39] suggested the importance of taking into account the full set of behaviours through real-time measurements to create models over time and, in consequence, infer patterns, context, and states of patients, with the last objective of developing personalised interventions. However, modelling methodologies rely on predictive strategies rather than the evolution of patient measurements. After this analysis, there is a necessity to advance towards a temporal and dynamic data-driven to succeed with the precision medicine paradigm [18].

Process Mining [40] solutions can offer a better understanding of a care process than other techniques used in previous works. Process Mining techniques are based on syntactical data mining framework thought to support experts in the understanding of complex processes, in a comprehensive, objective and exploratory ways [40]. Health processes are structured multidisciplinary care protocols and plans which detail essential steps in the care of patients within a specific clinical problem [22]. In this line, care pathways are complex processes including each stage of the management of a patient with a specific condition over a given time period, and include progress and outcome details. In that way, care pathways should be understood as a patient’s overall journey, instead of isolated functions independently.

The use of Process Mining can help obtaining individual healthcare processes. Process Mining provides algorithms, tools and methodologies to demonstrate what is actually happening within a process [41]. One of the main reasons Process Mining is being introduced in healthcare, is because it prioritises human understandably over accuracy. In consequence, Process Mining can be used to obtain knowledge from health information and comprehend dynamic healthcare processes. One of the main objectives of Process Mining is to infer knowledge from data, understanding data as recorded event logs, where each event refers to a case, an activity, and a point of time, in order to discover, monitor and improve real processes. All of this is done through three different ways: discovery, conformance and extension [42]. The application of Process Mining technologies can be used to support health professionals in the discovery of health processes and patients’ behaviours. Although the use of Process Mining in healthcare is emergent, several works have shown the feasibility of applying Process Mining in this domain [43]. Process Mining techniques have been used for administrative analysis in health domains for specific illnesses [44], like gynaecological oncology [45], for the analysis of services like Emergencies [46], or even for modelling the human behaviour [33]. In Reference [47], there is a first attempt to characterise processes for the patient’s conduct with promising results. Similarly, Reference [48] has approached the analysis of user behaviour using Process Discovery techniques to derive activity models from sensor activation logs in a smart environment; or Reference [49], which proposes obesity processes’ characterisation with PM techniques.

## 3. Materials And Methods

### 3.1. Data Source

Ideally, data would be generated from different and specific smart sensors, such as a WiFi scale for weight (BMI), a blood pressure monitor, and/or a blood glucose device. However, the adoption of these sensors is not widely extended across the population, and they are not connected to public health services or application we can use. In consequence, retrospective data from EHR was used as a proof of concept, to demonstrate the validity of the approach proposed in this paper. Data coming from sensors and EHR have many similarities as they provide the same result data, however they also have some differences regarding data frequency and availability. Finally, this study was conducted using data directly extracted from the database of a tertiary hospital in Spain, in a retrospective manner from 2012 to 2017, from 50,196 unique patients as described in Table 1.

Concretely, the used database contained data from primary care service, emergency, outpatient, and morbidity diagnosis service, as described in Table 2. All data were anonymised prior to the extraction.

### 3.2. Interactive Process Mining

Process Mining can provide a solution for Data Driven discovery of dynamic risk models. Risk Values of individuals can be seen as events of the patient behavioural risk process. With these events, we can create Process Mining Logs that can be used to discover the flow followed by a risk model in the patient’s care process. With these views, health professionals can not only better understand behaviours and risk models, but also, they allow the extraction of evidence based on the correlation of the risk dynamic behaviour and the adverse outcomes suffered by the patient. So far, it is also important to highlight the importance of experts in the healthcare process, within this, the Interactive Pattern Recognition (IPR) [50] is a formal framework, which introduces the health expert in the learning process and allows him/her to correct the hypothesis model in each iteration to prevent unsatisfactory errors and to assemble to a solution in an iterative way [51]. Within an interactive paradigm, the professional could acquire knowledge and understanding about what is actually happening during a period, instead of visualising isolated data of a single individual that do not provide extra insight from the health process. This methodology can be used to apply precision medicine in a more individualised way, supporting experts in the evolution of models in parallel to the evolution of patients’ behaviour.

### 3.3. Methodology

The main objective of the work is to build three Interactive Process Indicators for the three chronic conditions considered, using the three metabolic risk factors associated. For this, we have used an interactive methodology based on the Interactive Process Mining paradigm [51], using the solutions provided by PMApp tool. PMApp is a Process Mining toolkit that is based on the PALIA Suite tool [46]. PMApp enables the creation of custom Interactive Process indicators specific to the Medical Domain. The followed methodology is showed in Figure 1 and implements six main steps to obtain the corresponding IPI.

The methodology starts with the data ingestion, where the Data Log is obtained with the appropriate format to perform the Process Mining itself. In the second step, data are processed to compute the needed variables to create the events and traces for the PM analysis. After applying the filtering and processing step, the Log is ready for obtaining the Process Model behind the data using the appropriate discovery algorithm. PMApp tool provides PALIA (Parallel Activity Log Inference Algorithm) as discovery algorithm [52]. PALIA has been widely tested in real healthcare scenarios. It has been applied to the analysis of follow up protocols of patients with diabetes [35,53]; to measure and discover the individualised behaviour of older adults at risk of dementia [33]; for the characterisation of emergency flows, measuring organisational changes effects [46], for discovering surgery department flow [52], malnutrition assessment [49] or obesity characterisation [54]. PMApp also enables the creation of interactive dashboards that respond to the selection of arrows and nodes by capturing GUI events and it also allows the user to create custom forms and algorithms for discovery, filters, enhancement maps, and so forth [46]. On the other hand, to allow stratification based on the behavioural aspects of risk models, it is necessary to detect patients with different risk behaviours. Clustering algorithms are unsupervised data mining solutions that are able to group traces that have similar behaviour, maximising differences with the rest of groups. Process Mining in combination with Trace Clustering techniques can be a solution for that problem [55] and the PMApp tool also incorporates this possibility.

After the discovery, the next step is comprised of the computation of the metadata associated with the model. PMApp supports metadata correlated to models in several ways, such as statistical information associated to nodes and transitions, or the relationships between the topological structures of the model and the log events. Until this point of the methodology, the focused was on the accessing, collection, and processing data, but it remains one of the main steps, which is to present information to health experts. In PMApp, it is possible to render maps that can enhance the discovered model using colour gradients. With this feature, it is possible to render specific maps that highlight specific situations that depend on a customised formulation and are represented by nodes. This technique can be used to facilitate the health professionals’ understanding of the processes. Maps for common aspects can be created, such as performance, duration of activities, the number of cases, the number of events, and so forth. Moreover, specific maps that highlight specific situations customised for the problem can also be conceived. This can be used to show more specific representations that provide medical doctors with more personalised support to increase the usability, utility and reliability of the technology. The result is an IPI, a graphical model of a disease taking into consideration the evolution in a understandable way.

## 4. Results

As stated in Section 1, in this work we demonstrate the possibilities of using smart sensors joint with Process Mining techniques for the better management of chronic conditions, specifically some of the most prevalent non-communicable diseases, they are obesity, high blood pressure (hypertension) and hyperglycemia (Diabetes). Using the Interactive Process Mining methodology, dynamic models associated with these chronic diseases were obtained as Interactive Process Indicators for the understanding, measurement, and optimisation of the processes associated with obesity, hypertension, hyperglycemia, allowing health professionals to navigate behind the models and to discover the specificity of the processes correlated with individuals.

As explained in Section 3, the methodology starts with the data ingestion. The hospital experts provided the data in several Comma-Separated Values (CSV) files, concretely one CSV file per table included in Table 2, where values were represented in a set of rows and columns. At this point, it was performed the selection of the relevant data for the creation of the corresponding IPIs. Taking into consideration the chronic diseases under study, obesity, hypertension and hyperglycemia, information was extracted from *Patients Anonymize*, *Primary Care* and *Laboratory*, a description of which is included in Table 3, Table 4 and Table 5 respectively.

At this stage, data were processed to compute the variables that are needed to create the events and trace data for the PM analysis. For this, two actions were carried out—format corrections and the addition of new semantic values. Format corrections were applied to Measure Date, Test Request Date, and Numerical Results. A semantic result provides a semantic vision that facilitates the understanding of the chronic condition process semantically, this means to associated a semantic value to a numeric one. These values are disease depending, so semantic results were added for the three conditions under study. In the case of obesity, and following WHO recommendations [10], the BMI semantic result was introduced as follows—*Underweight* for BMI numerical result less than 18.5; *Normal* for BMI between 18.5–24.9; *Overweight* for BMI between 25.0–29.9; and *Obese* for BMI greater than 30. For hypertension semantic results, there were considered the cut-off points specified by the American Heart Association (AHA) [56]. So semantic results were *Normal* for SBP numerical result < 120 mmHg and DBP numerical result < 80 mmHg; *Elevated* for SBP between 120–129 mmHg and DBP < 80 mmHg; *Hypertension stage 1* for SBP between 130–139 mmHg or DBP 80–89 mmHg; and *Hypertension stage 2* for SBP ≥ 140 mmHg or DBP ≥ 90 mmHg. Finally, in the case of hyperglycemia, the measurement of glucose in the blood remains the mainstay of testing for glucose tolerance status, this could be obtained by laboratory measures and nowadays by portable devices. We followed the current WHO diagnostic criteria for diabetes type 2 [12,57]. The Diabetes semantic results were *Diabetes* for FPG ≥ 126 mg/dL; *Intermediate Hyperglycemia* for values of FPG between 100–125 mg/dL; and *Normal* for FPG less than 100 mg/dL.

Event data were composed of a *Start* corresponding to the field *Measure Date*; the completion time or end adding a second to the start; the name of the node, the identification of the trace, and the metadata correlated with the event. The name of the node was based on the semantic results as Named events, defined by the clinicians according to the mapping of the process. The identification of the trace corresponded with the *ID_ANON*. Whereas the trace data, considered as the set of metadata related to the same case, included the *Age Group*. At this point, the Process Mining Log was created and ready for the next stage, filtering, and processing the data to select the adequate Log for constructing the appropriate IPI. From here, different filtering strategies were followed for each IPI, consequently the rest of the process is particularised for each condition.

In the obesity case, five different filters were implemented and applied in a concrete order. Void traces were deleted, there were selected patients with more than four observations during the period, then traces were sequenced assuming ending of the current trace was the beginning of the next one, and finally a fuse filter was applied to merge equal traces. At this point, from the 17,853 initial unique patients, there were obtained the flows for 2260 patients after implementing previous filters.

Ultimately, a clustering filter was used for stratifying the population with similar behaviour based on the semantic value of BMI, extracting sub-logs from the main log. We have selected Topological Distance as it maximises the similarity between two traces, concretely *Weighted Topological Distance (WTD)* [33] augments similarity in the topology structures of the inferred workflow. This distance was used with Quality Threshold Cluster (QTC) [58] as the Clustering algorithm. QTC algorithm requires a *quality threshold* to decide the maximum distance among traces in the cluster. At this point, the best results arose with a quality threshold of 0.12 for the clustering algorithm and 0.01 of similarity.

A similar strategy was applied in the case of Hypertension but implementing six different filters in a specific order. Void traces were deleted, only patients with both, SBP and DBP, measures at the same moment were selected, and with more than four measures during the period. Then, traces were sequenced assuming the ending of the current trace was the beginning of the next one, and equal traces were fused. After filtering, from the 17,853 initial unique patients, we have obtained the flows for 3575 subjects. Likewise, trace clustering using WTD and QTC was used to obtain sub-populations based on BP behaviour using the semantic results. In this experiment, the best results were achieved for the quality threshold of 0.15 and 0.02 similarity.

Finally, for the hyperglycemia case, a similar approach was performed, implementing four filters. Due to the elevated number of measures included in the *Laboratory* table, we have extracted measures for FPG and glycated haemoglobin (HbA1c), discarding the rest of the data. Then, as in the other models, we deleted traces with void results or measures, and only patients with three or more observations during the period were used in the analysis. Last, traces were sequenced assuming ending of the current trace was the beginning of the next one. From the initial 50,196 unique patients with 18,182,239 observations from the *Laboratory* table, there were discovered the flows for 25,992 patients with 328,545 observations after the process of filtering and processing.

Following the methodology schema (Figure 1), after this point the Log was ready for obtaining the Process Model behind the data, using the appropriate discovery algorithm. After applying PALIA we obtained the Process Model ready to be processed in the next step. To create useful dynamic models of chronic conditions is needed to compute the metadata related to the model, for example, two patients could have the same BMI events, *Overweight*, but their timing and frequency could be completely different. It is crucial to analyse these differences in the process of understanding the dynamic characteristics of a model. For this reason, after applying the discovery algorithm, we have processed the log obtained to compute the metadata associated with the model. PALIA supports metadata correlated to models in several ways, concretely in this work we have used metadata computed to nodes and edges with statistical information, so we can appreciate how the executions of the models have been performed. This statistical information contains the execution number, the duration average, the duration median, the duration aggregation, the case number, and the duration by case.

Until this point of the experimentation’s approach, we had been accessing, collecting, and processing data, but it remains one of the most important steps, which is to present this information to the health expert in the form of the three Interactive Process Indicators for obesity, hypertension, and hyperglycemia.

### 4.1. Dynamic Characterisation of Obesity

As said, the result from the previous experimentation in the case of BMI represents the model for the *Dynamic Characterisation of Obesity* behaviour of the population considered. The model characterises the population into nine sub-population groups, showing different weight behaviours, these groups are listed in the Table 6.

These nine groups, plus the outliers, are included in the figures gathered in Figure 3, from the most prevalent dynamic behaviour represented by the obese population (cluster 0) to the less prevalent observed by the normal BMI population (cluster 8). Models have been enhanced by a heat map, where, the nodes, have been coloured with a gradient that means the median time of stay, and edges have been painted with a gradient symbolising the number of patients, that, proportionally follow this transition, where gradient scale goes from green (minimum value) to red (maximum value), as represented in Figure 2.

The nine models for obesity characterise the considered population into well-defined categories regarding their weight behaviour, based not only on BMI status but also and more important on individuals’ evolution and behaviour. These behaviours include common patterns of population with stable BMI, whatever it is, this is the case of clusters 0 (Figure 3a), cluster 1 (Figure 3b) and cluster 8 (Figure 3i). The model also shows a population with increasing BMI patterns, where a clear weight gain is shown in the behaviour, this is the case of cluster 2 (Figure 3c) and cluster 3 (Figure 3d). In the same way, decreasing BMI patterns are also represented in the model, concretely in cluster 4 (Figure 3e) and cluster 6 (Figure 3g). Finally, two more sub-populations have been discovered, representing unusual BMI patterns. This is the case for cluster 5 (Figure 3f), which includes obese population moving from the underweight state and going back to the initial situation, in a very short period (less than three months). After analysing this concrete group with health professionals, they indicated measurement errors as the most plausible explanation for this behaviour. Similarly, cluster 7 (Figure 3h) includes some unusual weight changes, moving from an overweight situation to an underweight state that could be explained by special situations such as surgeries or pregnancy.

### 4.2. Dynamic Characterisation of Hypertension

The IPI obtained representing the model of *Dynamic Characterisation of Hypertension* was obtained after applying the explained Interactive Process Mining methodology. The model describes the population regarding their dynamic blood pressure behaviour. On this occasion, there were 13 groups obtained with different blood pressure flows that are included in the Table 7.

These groups are included in Figure 4, from most prevalent to less prevalent, respectively. Models were also coloured with a gradient for nodes by the median time of stay, and edges have been painted with a gradient symbolising the number of patients, that, proportionally follow this transition, where gradient scale goes from green (minimum value) to red (maximum value), using again the gradient scale represented in Figure 2.

These results have characterised the population under study into 13 well-defined groups regarding their dynamic evolution of the BP. These groups show intrinsic variability of blood pressure, as blood pressure is a continuous variable that fluctuates in response to various physical and mental changes. Considering the BP as a dynamic process, this variability is included within the different models. Some dynamic patterns are found within the model. This is the case of the population with dynamic increasing patterns for BP, these patterns are represented in cluster 2 (Figure 4c) with patients moving from elevated and hypertension 1 to hypertension stage 2; and cluster 11 (Figure 4l) showing a similar behaviour, moving from normal BP to elevated BP or hypertension stage 2. Similarly, decreasing patterns are also discovered and represented in cluster 12 where most of the patients finalised the period with normal BP although they came from other stages with elevated BP (Figure 4m); cluster 6 shows decreasing pattern from hypertension stage 2 to elevated BP (Figure 4g); and cluster 10 (Figure 4k). Stable BP patterns are also identified for normal BP (Figure 4b), hypertension stage 1 (Figure 4f), and hypertension stage 2 (Figure 4i). Finally, irregular patterns include patients with constant changes in their BP values, showing they have not controlled their BP. These groups are: cluster 0 with patients changing between hypertension stage 1 and 2 (Figure 4a); cluster 3 shows patients finalising with normal BP but with long episodes of hypertension illustrating how important is consider the whole process (Figure 4d). Cluster 4 (Figure 4e), cluster 7 (Figure 4h) and cluster 9 (Figure 4j), all are clear examples of patients with decompensated BP with several episodes of hypertension.

### 4.3. Dynamic Characterisation of Hyperglycemia

The third obtained IPI corresponds with the model of *Dynamic Characterisation of Hyperglycemia*. The purpose of this IPI is to characterise the diabetes type II behaviour of the studied population trough the FPG flow, thanks to the Interactive Process Mining methodology. The results are presented in Figure 5 in which, as previous IPIs, nodes are coloured by the average time spent in the stage, and edges have been painted with a gradient symbolising the number of patients, that, proportionally follow this transition, from green (minimum value) to red (maximum value), another time using the gradient scale represented in Figure 2 for nodes and transitions, respectively.

When considering all the population, the *Normal* stage for FPG is the most prevalent on average, but population also spent a considerable time in *Intermediate Hyperglycemia* and *Diabetes* stages. It is not only important the time spent in each stage but also the transitions among stages, as this can suppose the difference between a well-controlled glucose status or not. The reddest transitions correspond with the number of patients, that proportionally follow this path, and in this case Normal to Diabetes is the most followed one. However, as explained in Section 1, some factors such as age, gender, and ethnicity, are relevant for stating normal glucose level. Age is available in the database used, so it was feasible to analyse how it affects FPG flows and to find more relevant views for health experts. For that, we divided population in three main groups of age, *Young* from 20–30, *Adults* from 30–65, and *Elderly* from 65–100. FPG behaviour for young, adults and older adults are included in Figure 6, Figure 7 and Figure 8 respectively.

With decreasing age, we can observe how time spent in *Normal* stage increases, with the corresponding decrease in the time consumed in the *Intermediate Hyperglycemia* and *Diabetes* stages. We can also notice the most prevalent path is the one in which the population finalises in the *Diabetes* stage. A more in-depth analysis can be done using enhancement possibilities. The IPI can be enriched with a map highlighting the differences between the process nodes and edges and their degree. Figure 9, Figure 10 and Figure 11 include the IPI with enhancement showing negative differences in red colour, where the saturation of the colour reflects the degree of differences in negative (reddest) or positive respectively, in this case using the gradient scale showed in Figure 12 from white to red, both for nodes and transitions.

Differences envisaged with the observation of the three age cohorts are shown within this enrich IPI. The comparison of the three groups with the total population shows a negative difference in Intermediate Hyperglycemia and Diabetes nodes, the degree of the difference is greater when age decreases. In addition, a negative difference is observed in transitions between Intermediate Hyperglycemia and Diabetes nodes, but also between Normal and Intermediate Hyperglycemia, this difference increases in the young population (Figure 11).

The comparison of groups can also help health professionals to discover their differences and this can allow them to understand group characteristics. In medicine, a classic trust measure to evaluate and measure medical processes is to show differences among them, that is known as statistical significance. Most of the literature focuses on the *p*-Value for measuring the statistical significance [59]. PMApp implements the statistical significance using *p*-Value comparing nodes that refer to the same activity. Then for each execution associated with each activity is got the set of times and applied the Kolmogorov-Smirnov Test in order to evaluate the normality of the distribution of the time values. At this moment, if the two distributions reach the normality test, then it is used a T-student Test for the *p*-Value computation. If not, it is assumed the distributions are not normal and the Mann-Whitney-Wilcoxon Test is performed. If both situations, for a *p*-Value lower than a given threshold, it is concluded that the distributions are significantly different. Following the literature, the threshold was set to 0.05 [46].

This technique can be used to highlight the differences with statistical significance between the two models referring to two cohorts. This approach can not only discover when a process is different but also in which parts of the models the differences lie [46]. Figure 13 and Figure 14 show where processes for adult and young populations differ with respect to the elderly process. Nodes highlighted in yellow mean that there is a statistically significant difference between the Adult and Elderly cohorts, and the Young adults and Elderly cohorts respectively. For example in Figure 13, it can be observed that elderly population spend significantly more time in the *Diabetes* stage, whereas they spend less time in the *Normal* stage. Comparing young population with respect to the elderly one, Figure 14, these differences are even more significant statistically speaking, as the young population substantially consume less time in the *Diabetes* and *Intermediate Hyperglycemia* stages, and more time in the *Normal* state. With these results, we can confirm how age is affecting the FPG flow in the average time spent in each state of their processes.

## 5. Discussion and Conclusions

With retrospective data from a tertiary hospital (include in Table 2), we have demonstrated how the Interactive Process Mining methodology could be applied as a new analytical method for the better handling of chronic conditions using data coming from patients over a period. This has been approached with the analysis of three common and prevalent chronic diseases—obesity, hypertension, and hyperglycemia. As a proof of concept of the overall strategy, and due to the impossibility of integrating data coming from sensors of a considerable sector of the population, we have used real data from a Health Electronic Record of a tertiary hospital.

In this concrete scenario, we have obtained three valuable and innovative Interactive Process Indicators that could be used for understanding, measuring, and managing the processes for the three underlying conditions. These three IPIs have the potentiality of presenting findings over data as comprehensible insight views, with the ultimate goal of health experts could discover new medical evidence.

Although in this study we have worked with data collected from an EHR, this can be self-adapted and automated to a population using personal devices collecting the same parameters or even more. This strategy, in combination with smart sensors and personal devices, could allow health professionals to analyse individual behaviours and to compare current behaviour with part of the inferred workflows or with other cohorts, and to measure changes in treatments and adherence.

Considering the three studied chronic conditions, the Interactive Process Mining methodology has permitted to characterise the population in a dynamic and personalised way for the three conditions. In the first case, the IPI *Dynamic Characterisation of Obesity* has discovered nine sub-populations with well-defined BMI patterns. Three evolution patterns were discovered in the models, one pattern for patients with a stable weight, but two other groups that change their weight, with increasing and decreasing patterns. This finding is very relevant, as we have been able to stratify the population based on their weight evolution, we were even capable to detect measure errors, and this will permit to treat them in consequence. If we consider two patients with the same BMI, but from two different risk models, the first one from the stable overweight pattern (Figure 3b) and the second one from a decreasing pattern, for example within cluster 4 (Figure 3e); they have the same BMI at the end of the period, overweight, but their behaviours are clearly different. In a classic and static approach, the only insight is the BMI result or ’number’, however, the IPI view lets us considering other dimensions of the problem. For example, the first patient has not made any improvement in her/his health status at any moment, therefore the patient is probably not well-engaged with diet counselling or not properly motivated. On the other hand, the second patient is losing weight, she or he is doing things well and treatment is working. In consequence, disease management should not be the same for these two patients, and personalised interventions should be delivered in order to succeed with weight loss. In the first case, health professionals could influence general health behavioural changes, whereas in the second case they could continue motivating the patient to maximise correct attitudes. This IPI has allowed the classification of the population regarding dynamic weight behaviour and has shown insights in an understandable way. With this information, health professionals could put in practice concrete and personalised interventions in specific groups trying to influence in particular behaviours. This result characterises a sample population with weight data from one year (see Table 2), in this case, the weight parameter comes from a hospital’s EHR, but the results could be analogue with data from a smart sensor at patients’ home or wearable. The strategy could not only be the same but also with more information from personal devices, a more complete and accurate model would be obtained.

In the case of hypertension, the IPI *Dynamic Characterisation of Hypertension* shows 13 different patterns with the continuum of BP and its evolution. Analysing this IPI, health professionals could compare the evolution of the BP in different groups, personalise interventions, and test their efficacy and effectiveness over time. As in the previous case, data come from real patient of a tertiary hospital during a year, however, the use of smart sensors could arise better results. The use of BP monitors at home by patients to collect BP measures, in combination with Interactive Process Mining methodology for their analysis, could provide a great opportunity not only to discover a more precise models but also to examine BP variability and relationship with other factors, such as the moment of the day when BP is measured, or the period of the year. Moreover, the combined analysis of BP with other parameters collected by personal devices, such as mood, activity, and so forth could open a new dimension to understand BP processes, its relationships with other conditions, and in consequence better management and interventions.

Finally, hyperglycemia has been modelled by the IPI *Dynamic Characterisation of Hyperglycemia* using FPG to evaluate the continuum and evolution of blood sugar management. In this case, we have used a larger sample, with data from more than seven years (see Table 2), with the possibility to establish differences among the population. At first sight, the model reveals that the time spent in diabetes and intermediate hyperglycemia stages is, on average, considerably high for the group considered. As the model takes into account the behaviour over time, instead of a concrete situation or the transition between two concrete values, it is possible to see the patient’s evolution. The model could show an issue of under-diagnosis and consequently under treatment, or a problem with the treatments and their follow-up, as the model insights a population that is not well-controlled regarding diabetes. Further analysis with health professionals is needed to determine what is the cause and properly react. In any case, if health professional have this information, they could take the appropriate decisions. Only when health professionals can analyse information behind data, reasonably, they will be able to treat patients accordingly. This result could be easily adopted in an scenario in which patient use a personal glucose monitor. The use of a smart sensor for blood glucose monitoring will enrich the database, and consequently the IPI discovered, not only with more measures of the considered factor but also with other personal and environmental information, valuable to obtain more precise and personalised models.

Moreover, the comparison of cohorts could also help health professionals to discover their differences and to appreciate their characteristics, where techniques such as enhancement help to discover when a process is different and in which parts of the models the differences lie. With maps that highlight the differences between the process nodes and edges and their degree, the expert could understand the process in a better way. With the selection of the the age variable to see differences in FPG flows, we have obtained an enriched IPI for Hyperglycemia, where cohorts for three different groups of age show a clear difference in the FPG flows. The average time spent in normal stage, that is associated with low risk of developing diabetes or cardiovascular disease, decreases with age, and in consequence, increases the risk of associated problems. Furthermore, it is not only an observed difference, this is statistically significant as the enrich IPI shows in Figure 13 and Figure 14. The process of building IPIs associated to glucose levels using new glucose sensors would move the field of this chronic disease to a new level of understanding, modelling diabetes in a dynamic and personalised manner, including factors as age, gender, ethnicity, socioeconomic factors or lifestyle in the process, to understand what is happening behind a glucose test result. This would let personalised medicine certainly implement adaptive and personalised treatments.

These IPIs suppose a step forward in the personalised medicine concept, incorporating evolution over time and patient’s unique behaviour to the analysis. Discovered IPIs for the three chronic conditions consider the variability over time of the risk factors associated, and diseases as a process, where the current state is as important as the previous ones, the evolution, and the time spent in each state. These models also consider stratification groups for common patterns, and the association with other factors, such as age, comorbidities, gender, and so forth and can incorporate data from different sources. These results, compared with other approaches that treat chronic conditions under a static and time-invariant set of concepts inferring linear relationships among variables, suppose a new opportunity to analyse data from other perspective. Although some techniques, such as Temporal Abstractions, have considered a knowledge-driven means, it is needed to implement a data-driven resolution so patients’ behaviour could be obtained from current data as a dynamic and temporal flow. This novel context includes the dynamic variability of the chronic conditions and the patients’ behaviour.

The results presented in this work, supposes for the authors the point of departure for a new promising model that enables extracting knowledge from data in the field of chronic diseases. Future work in this line should comprise the involvement of clinicians not only to validate the clinical utility of these results, but also to measure the validity in concrete patients. Working with the three studied chronic diseases, new measures could be added in order to enrich the models working in close collaboration with clinicians. On the other hand, authors envisage the development of dynamic models focused on other diseases or pathology, where the time and evolution perspective could suppose an added value. In this line new research could be done in the area of cancer treatment and post-treatment period, where the application of PM techniques to sensor data could be applied to individual patient ill-health trajectory modelling, visual exploration of interacting cancer symptoms and comorbidities signs, patient stratification, or quality of clinical cancer care service, combining clinical events from EHR and patients‘ response to treatment through sensors.

In this work, we have implemented an Interactive Process Mining strategy in order to build Interactive Process Indicators for obesity, hypertension and hyperglycemia diseases. This strategy let us go a step ahead in the area of data analytic, using data coming from population sensors we are able to create innovative models for chronic diseases, inferring real processes behind data. These IPIs take into account patient behaviour over time showing the variability of diseases and patients. Authors envisage the great impact this approach might suppose on health professionals practice, opening a new pathway to achieve personalised medicine. Next steps should be, on the one hand, engaging health professionals in the deep analysis of these results, only working in close collaboration with them would allow the improvement of current risk models and the infer of new ones. And on the other hand, to validate the hypothesis with data coming from sensors for weight, blood glucose and blood pressure, as main as establish other personal and environmental parameters to improve and enrich the IPIs.

## Figures and Tables

**Figure 1 sensors-20-05330-f001:**
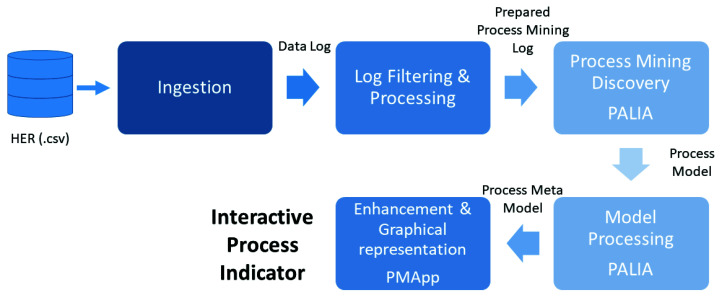
Methodology flow.

**Figure 2 sensors-20-05330-f002:**
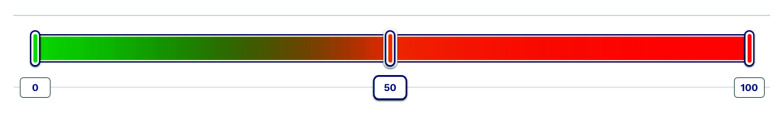
Gradient scale key for model representation from green to red.

**Figure 3 sensors-20-05330-f003:**
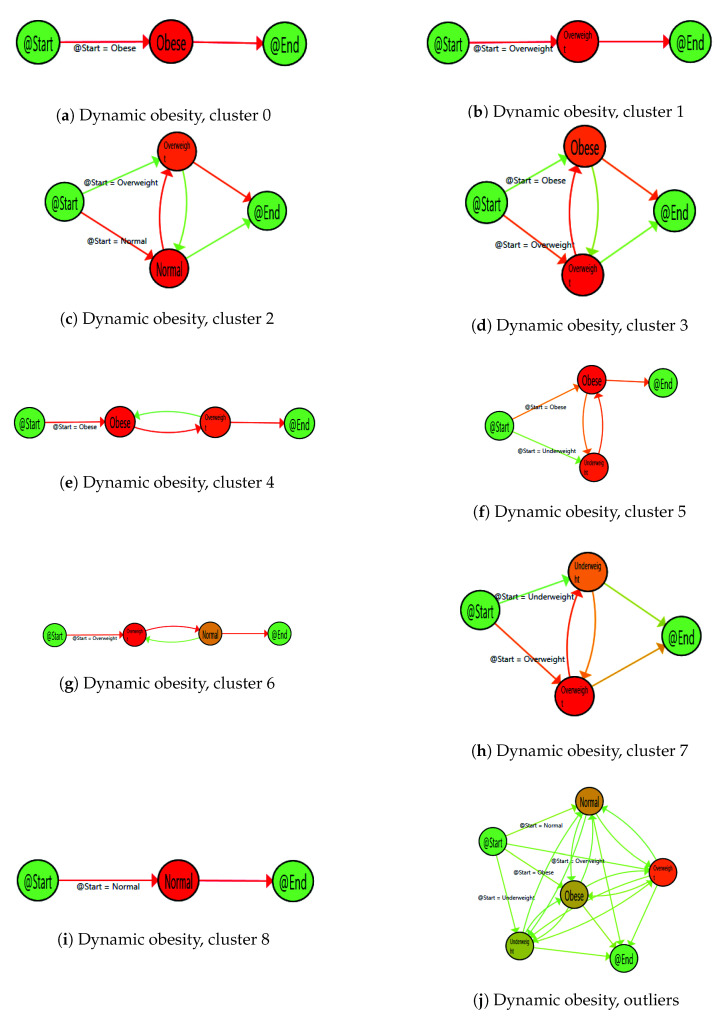
Dynamic characterisation of Obesity.

**Figure 4 sensors-20-05330-f004:**
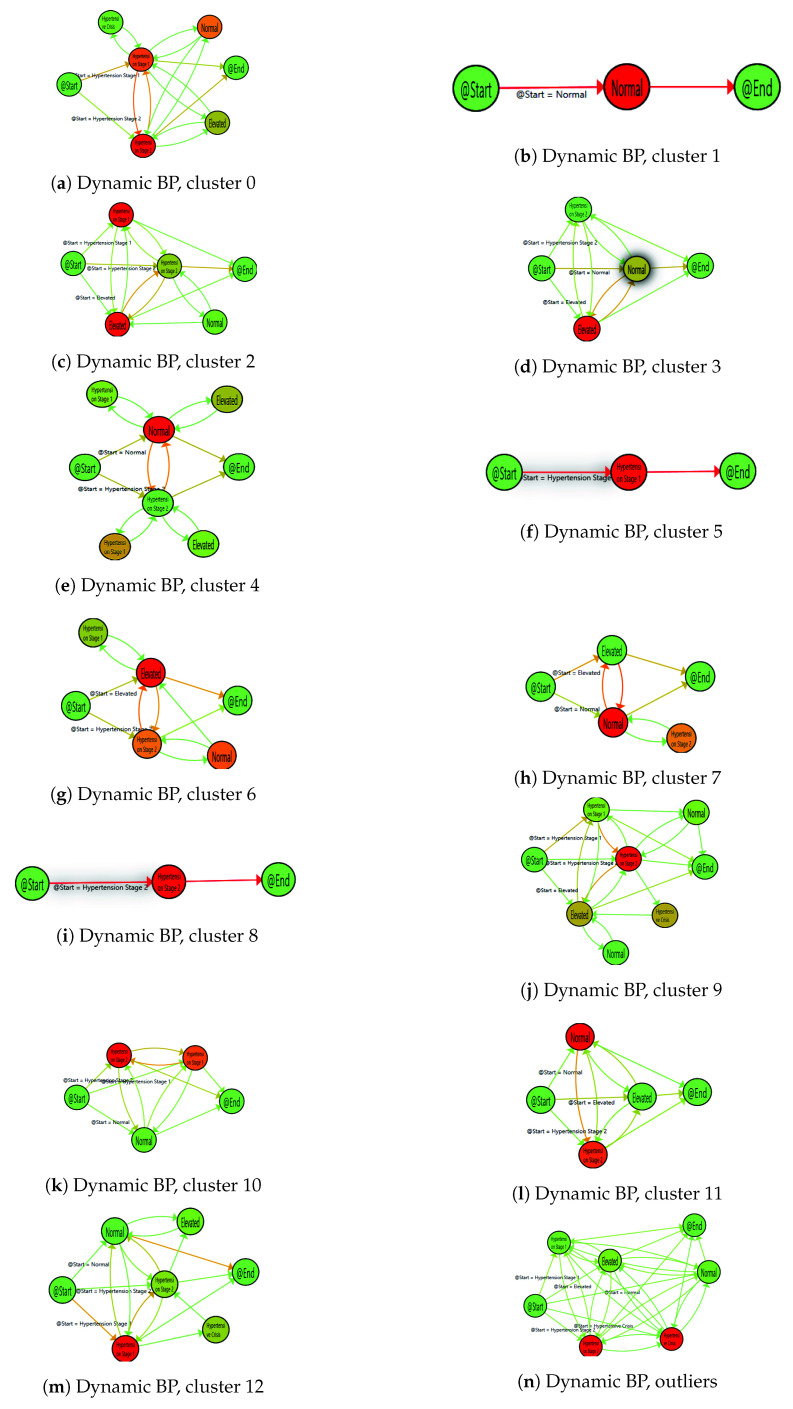
Dynamic characterisation of hypertension.

**Figure 5 sensors-20-05330-f005:**
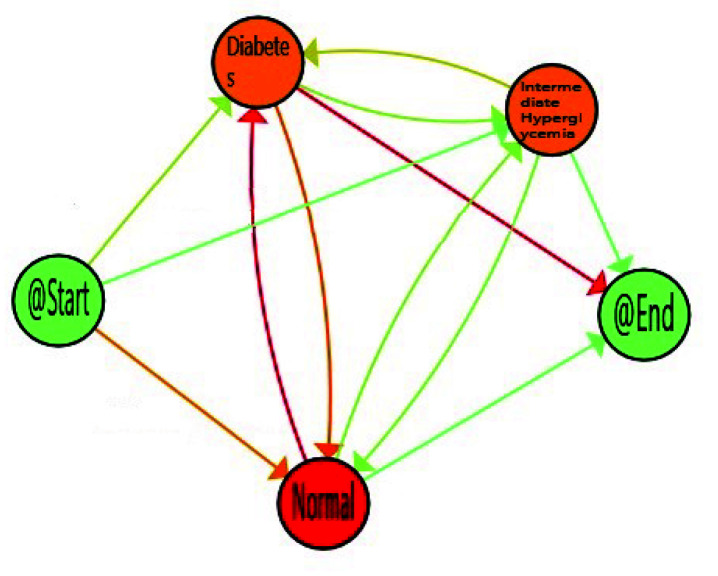
Dynamic fasting plasma glucose.

**Figure 6 sensors-20-05330-f006:**
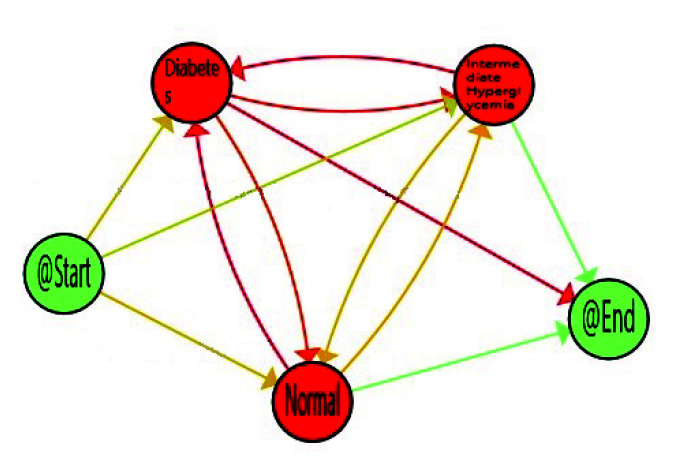
Dynamic fasting plasma glucose (FPG) for elderly (65–100).

**Figure 7 sensors-20-05330-f007:**
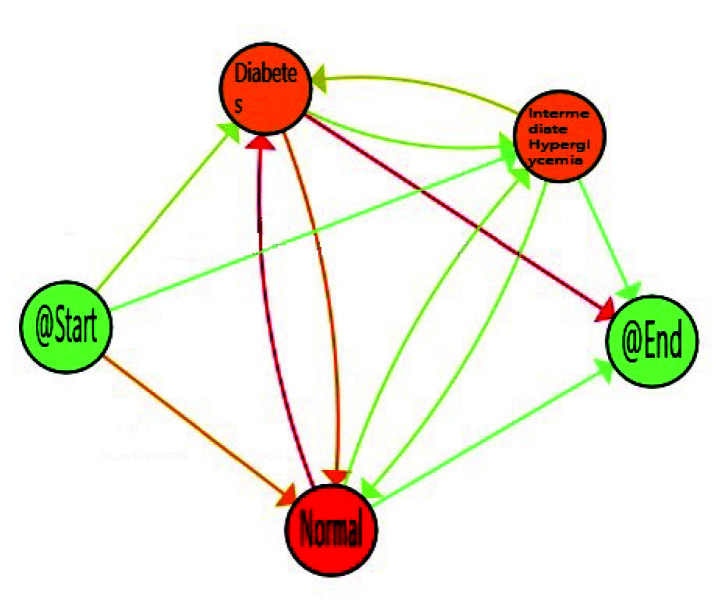
Dynamic FPG for adults (30–65).

**Figure 8 sensors-20-05330-f008:**
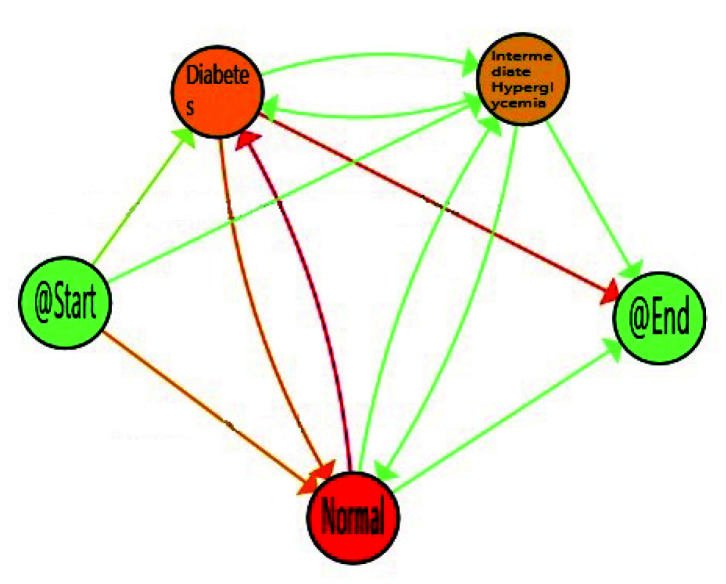
Dynamic FPG for young aduls (20–30).

**Figure 9 sensors-20-05330-f009:**
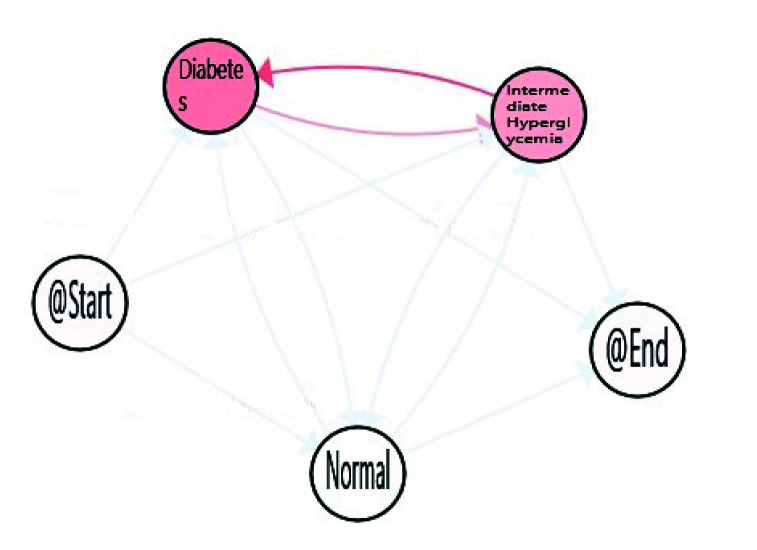
Difference enhancement model: All-Elderly.

**Figure 10 sensors-20-05330-f010:**
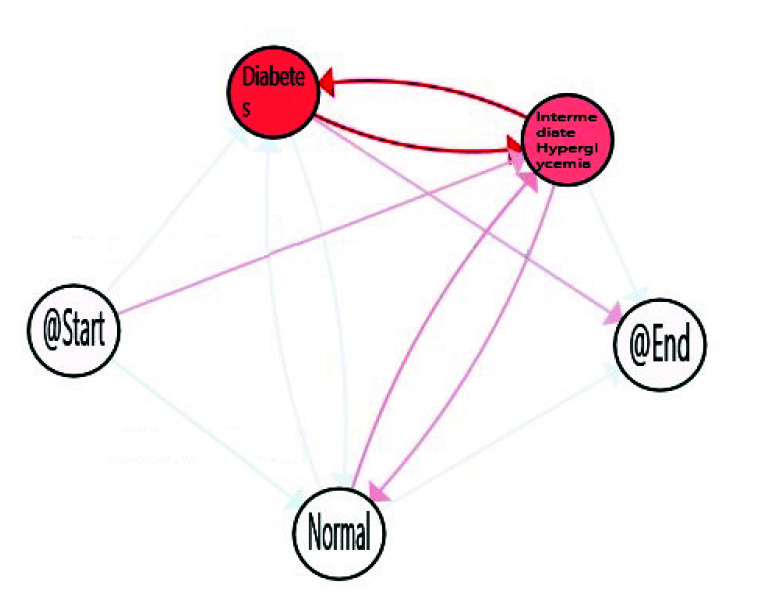
Difference enhancement model: Adult-Elderly.

**Figure 11 sensors-20-05330-f011:**
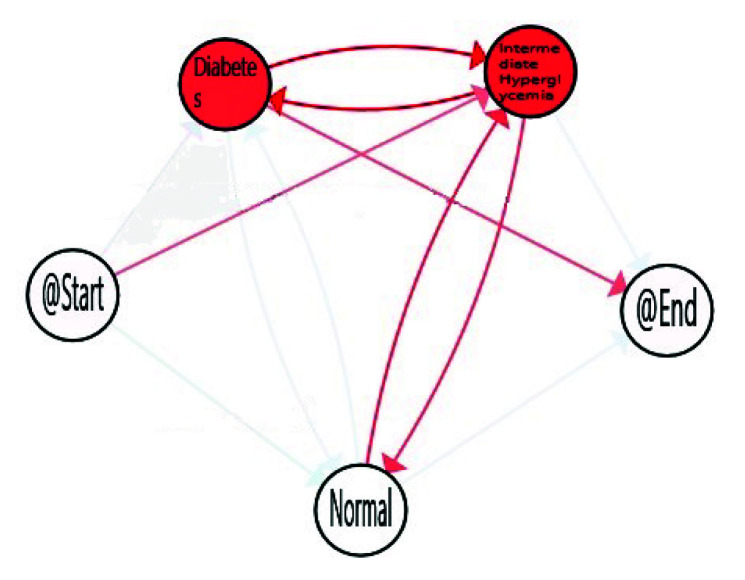
Difference enhancement model: Young-Elderly.

**Figure 12 sensors-20-05330-f012:**
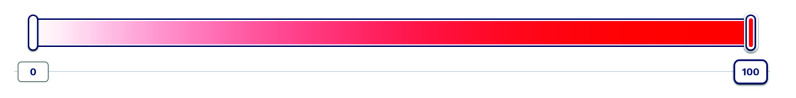
Gradient scale key for model representation from white to red.

**Figure 13 sensors-20-05330-f013:**
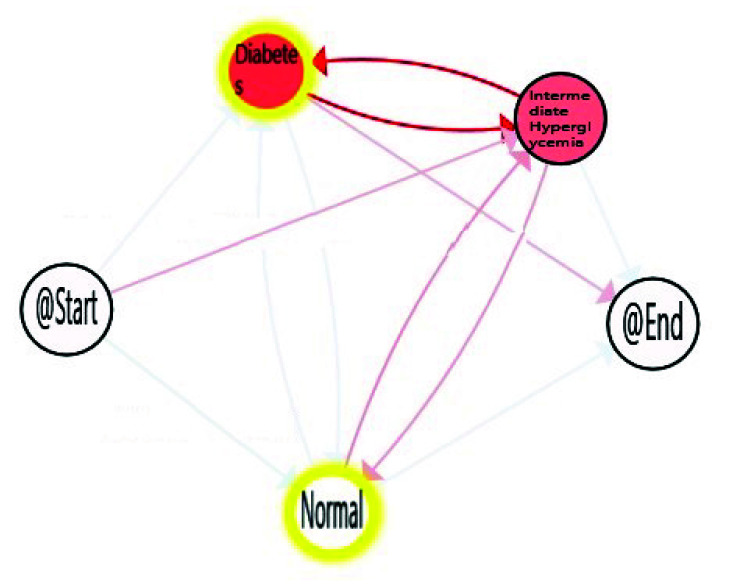
Differences between Adult-Elderly cohorts.

**Figure 14 sensors-20-05330-f014:**
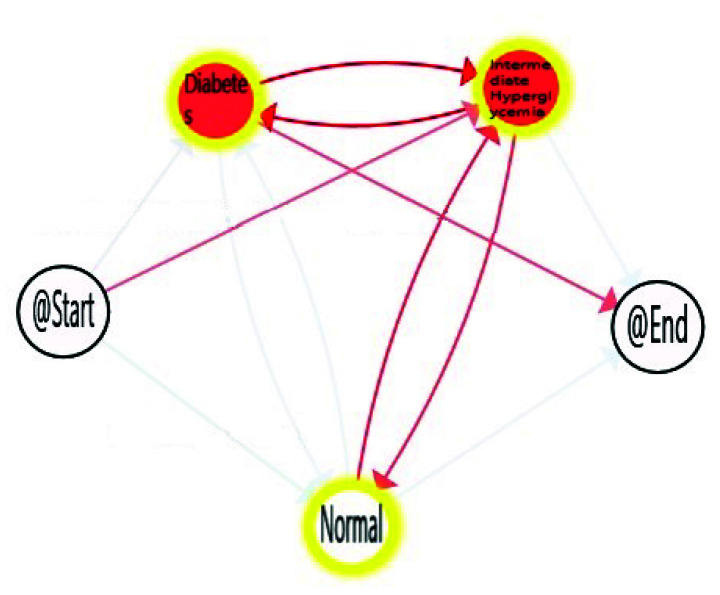
Differences between Young-Elderly cohorts.

**Table 1 sensors-20-05330-t001:** Data description.

Age Group	Population	Total %
15	498	1%
20	1838	3.66%
25	2075	4.13%
30	2752	5.48%
35	3919	7.81%
40	4209	8.39%
45	3821	7.61%
50	3692	7.36%
55	3499	6.97%
60	3509	6.99%
65	3879	7.73%
70	4345	8.66%
75	3699	7.37%
80	3381	6.74%
85	1967	3.92%
90	1193	2.38%
95	863	1.72%
100	521	1.04%
>100	536	1.07%

**Table 2 sensors-20-05330-t002:** Database description.

Table	Description	Unique Patients/ Observations	Period
Patients Anonymize	General information about patients: age, identifier, some diagnoses	50,196	-
Primary Care	Data collected in primary consultations: variables and annotations	17,853/215,523	2017
Hospital Admissions	Type of admission, ICD9 ${}^{a}$, Diagnostics, DRG ${}^{b}$, date	10,403/180,797	2012–2016
Emergency	Severity description, Admission service code, destination service, date	34,054/180,797	2010–2017
Outpatient	Provision type, date	6667/706,888	2012–2017
Morbidity Diagnoses	ICD9 ${}^{a}$ code, diagnose date	48,080/1,048,575	2012–2017
Laboratory	Laboratory measures: Date, id, description, result, units	50,196/18,182,239	2012–2017

${}^{a}$ International Statistical Classification and Related Health Problems, ${}^{b}$ Diagnosis-Related Group.

**Table 3 sensors-20-05330-t003:** Patients Anonymize table description.

Column Name	Data Type	Example
ID_ANON	Global unique identifier	000269d4-b40a-df4f-a1c0-56db3f989ad2
Age Group	Integer—group of age by 5 years	40
Overweight	Integer: 1/0, overweight diagnose	0
Obesity	Integer: 1/0, obesity diagnose	1
Unspecified Overweight or Obesity	Integer: 1/0	1

**Table 4 sensors-20-05330-t004:** Primary Care table description.

Column Name	Data Type	Example
ID_ANON	Global unique identifier	000269d4-b40a-df4f-a1c0-56db3f989ad2
Measure Date	String	20170830
Code Measurement	String—type of observation	BMI, Weight, Height, SBP, DBP,…
Numerical Result	Float—result of the measurement	87.5
Text Result	String—indicates void numerical result	Yes/No
Age Group	Integer—group of age by 5 years	45

**Table 5 sensors-20-05330-t005:** Laboratory table description.

Column Name	Data Type	Example
ID_ANON	Global unique identifier	000269d4-b40a-df4f-a1c0-56db3f989ad2
Test Request Date	String	20170830
Test Result Date	String	20170830
Test Id	Integer—test identifier	561
Test Description	String—measure description	Lipid index
Test Result	Float—test result	22.2
Test Units	String—code of the units	mg/dL
Age Group	Integer—group of age by 5 years	45

**Table 6 sensors-20-05330-t006:** Dynamic obesity sub-population groups.

Group Name	Population	Total
Cluster 0	742	32.8%
Cluster 1	683	30.2%
Cluster 2	269	11.9%
Cluster 3	204	9.1%
Cluster 4	105	4.6%
Cluster 5	57	2.5%
Cluster 6	53	2.3%
Cluster 7	47	2.1%
Cluster 8	40	1.8%
Outliers	60	2.7%

**Table 7 sensors-20-05330-t007:** Dynamic Hypertension sub-population groups.

Group Name	Population	Total
Cluster 0	810	22.7%
Cluster 1	335	9.4%
Cluster 2	310	8.7%
Cluster 3	290	8.1%
Cluster 4	275	7.7%
Cluster 5	185	5.2%
Cluster 6	159	4.4%
Cluster 7	154	4.3%
Cluster 8	118	3.3%
Cluster 9	110	3.1%
Cluster 10	108	3.0%
Cluster 11	94	2.6%
Cluster 12	82	2.3%
Outliers	545	15.2%

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
