# Peer review of "Dynamic Models Supporting Personalised Chronic Disease Management through Healthcare Sensors with Interactive Process Mining"

_sensors, 2020, doi:10.3390/s20185330_

Round 1

Reviewer 1 Report

This paper adopt the interactive process mining in medical data. The paper is interested and valuable for sensor data. Some improvements are suggested.

  1. Please explain the methodology (ex. a short introduction of the algorithm and applications)
  2. Please show the performance of results.
  3. List a table of comparisons about results.
  4. (Please see the attached file for details) 

Reviewer 2 Report

This paper discusses a technique for identifying dynamic risk models for chronic diseases on the basis of the patient's behavior analysis. The manuscript is identifying the results based on the dynamic characterization of the disease. There are few queries to be addressed such as,

  1. The authors are discussing so many models but what is their proposed model that is missing.
  2. Even though if the authors are executing data-driven discoveries of dynamic risk models, there must be some model description, however, we just observe the results and how these results are obtained that is not discussed in the form of any proposed model.

Reviewer 3 Report

This is a well presented and useful contribution at the intersection of process mining and healthcare sensor data. The authors present a solid overview of the relevant literature and explain their motivation with clarity. The scientific method is strong and the results interesting. There are a few areas where I think improvements could be made but these are largely presentational in nature: 

1) The case for using sensor-like result data from an EHR that has the same characteristics as sensor data was well made and it appears a pragmatic solution. There was some repetition of the case and it might have been better to make this case clearly at the beginning, discuss the similarities (e.g. same results data) and differences (e.g. greater volume/ velocity from wearables etc) in the study design section and a stronger discussion of the implications for the sensor community in the conclusion.

2) The use of PALIA and clustering to determine clinically relevant phenotypes is novel and exciting. The method is well described in detail but more general observations on the effectiveness of this approach, and alternatives (?) would be helpful. The presentation of the models resulting from the clusters should be improved ... consider a table with each of the clusters side by side (currently spread over several pages and disconnected from the text, especially Figure 9); what percentage and absolute numbers fall into these clusters; perhaps order them by most frequently found; add a key for the colours etc. 

3) The results are interesting but the next step would be to explore their clinical utility with real clinicians and perhaps for individual patients... does the identification of a patient within a specific cluster help the clinician understand how best to help the patient? The paper rightly states that this would be further work but a longer discussion on the nature of that further work would, I think, be a really helpful contribution to readers of this journal. 

4) Minor errors in English were found but should be corrected by careful proof reading. 

I enjoyed reading this paper and look forward to seeing it published. 

It would be good to see a statement on how ethical access to the confidential patient data was obtained. This can be included in acknowledgements or in the main text.   

Round 2

Reviewer 2 Report

The authors have incorporated the suggested changes in the article, and the manuscript seems a useful contribution to the healthcare society. Do make sure to check the typos in the paper.